# Regulation of Immune-Related Gene Expression by Salinity-Induced HPI Axis in Large Yellow Croaker, *Larimichthys crocea*

**DOI:** 10.3390/ijms26094298

**Published:** 2025-05-01

**Authors:** Jia Cheng, Zhengjia Lou, Huijie Feng, Yu Zhang, Honghui Li, Wuying Chu, Liangyi Xue

**Affiliations:** 1College of Marine Sciences, Ningbo University, Ningbo 315832, China; z20050693@ccsu.edu.cn (J.C.); NinaHu2013@126.com (Z.L.); Monica2023@yeah.net (H.F.); yzhangyu@163.com (Y.Z.); 2College of Biological and Chemical Engineering, Changsha University, Changsha 410022, China; chuwuying18@163.com

**Keywords:** *Larimichthys crocea*, brain transcriptome, salinity stress, HPI axis-related genes, immune-related genes

## Abstract

Large yellow croaker is one of the most popular economic fish species in China. There are studies on the effects of salinity on the growth and development of large yellow croaker (*Larimichthys crocea*), but the effects of the hypothalamic–pituitary–interrenal axis (HPI), HPI axis-related genes, and immune-related gene expression and its mechanisms have not been reported. This study analyzed the comparative transcriptomics of brain tissue in large yellow croaker under different salinity (12, 24, and 36 ppt) treatments for 4 weeks. The results showed that there were 1568 differential expression genes in the high salinity (HB) and normal salinity (NB) groups, including 494 up-regulated and 1074 down-regulated transcripts, and 1720 differential expression genes in the low salinity (LB) and normal salinity (NB) groups, including 486 up-regulated and 1234 down-regulated transcripts. Some pathways were significantly enriched, including the adrenergic signaling pathway of cardiomyocytes, oxidative phosphorylation, aldosterone synthesis and secretion, chemokine signaling pathway, and cyclic adenosine monophosphate (cAMP) signaling pathway. Quantitative Real-time polymerase chain reaction (qPCR) analysis further confirmed changes in the expression levels of HPI axis-related genes (*β2-ADR*, *GH*, and *PRL*) and significant changes in the expression levels of immune-related genes (*IL6st*, *IL6*, *CXCL12*, *CD40*, *IFNAR1*, *SOCS2*, *SOCS6*, and *IRF1*). In summary, this experiment demonstrates that salinity stress can activate the HPI axis and influence its immune function in large yellow croaker. Furthermore, the expression of immune factors during the immune response is regulated by the upstream genes of the HPI axis. The findings of this study are significant for understanding the physiological and immune responses of large yellow croaker to salinity stress.

## 1. Introduction

The stress response of fish is generally divided into a primary stage and a secondary stage. The primary stage is the body’s neurological and endocrine response, mainly to identify the source of stress and the activation of the biological defense system [1]. Two stress axes, the hypothalamic–sympathetic–chromaffin cell axis (HSC) and hypothalamus–pituitary–interrenal axis (HPI), take a role in the primary stage of the stress response in fish. The secondary stage mainly regulates the changes in tissue function, tissue structure, and energy metabolism caused by the primary stage, mainly in the changes in biological functions such as the respiratory system, energy metabolism, electrolyte balance, and immune system [2]. The activation of the HSC axis can cause the chromaffin cells in various tissues (mainly the head kidney) to release a large amount of catecholamines, mainly epinephrine and norepinephrine [3]. The HPI axis mainly uses the hormone cascade controlled by the hypothalamus to sequentially release the hypothalamic corticotropin-releasing factor (*CRF*), adrenocorticotropic hormone (*ACTH*), and interrenal tissue corticosteroids, eventually leading to a significant increase in plasma cortisol [4]. The β2-adrenergic receptor (*β2-ADR*), growth hormone (*GH*), and prolactin (*PRL*) are key factors in the HPI axis and play an important role in neuroendocrine. *β2-ADR* can achieve immune regulation and mediate the body’s immune response function by binding to catecholamines [5]. *GH*, as a polypeptide hormone secreted by pituitary eosinophils, plays a key role in the growth and development of bony fish and the regulation of osmotic pressure and is directly regulated by the hypothalamus [6]. *PRL*, as a multifunctional hormone with immune-stimulating effects, is not only indispensable in maintaining homeostasis but also plays a very important role in the immune–neuroendocrine regulatory network of the organisms [7].

Salinity is one of the important factors influencing fish survival and growth. Salinity changes can cause adaptive changes in gene expression in fish [8]. When the rainbow trout (*Oncorhynchus mykiss*) are transferred from freshwater to seawater, the messenger ribonucleic acid (mRNA) expression levels of corticotropin-releasing hormone (*CRH*) and corticotropin (*ACTHα* and *ACTHβ*) in the HPI axis genes increase significantly [9]. Plasma cortisol levels of gilthead seabream (*Sparus aurata*) transferred either from seawater to hypersaline water or from seawater to low salinity water increase on the first day [6]. The levels of *GH* and insulin-like growth factor 1 (*IGF-1*) mRNA were significantly increased in juveniles of *Argyrosomus regius* when the salinity changed [10]. When *Sparus aurata* are transferred from a salinity of 20 g/L to 38 g/L, the mRNA expression level of the *PRL* gene decreased significantly [11].

Salinity also affects the expression of immune genes in fish. For example, high salinity stress can cause the down-regulation of toll-like receptor 2 (*TLR2*) and interleukin 1 receptor type II (*IL1R2*) expression in Japanese eel (*Anguilla japonica*) [12]. In a study of Nile tilapia (*Oreochromis niloticus*), the mRNA transcription levels of interleukin-1β (*IL-1β*) and interferon-γ (IFN-γ) genes were significantly lower in the vaccinated fish at 20 and 30 ppt salinity, as compared to fish at 0 and 10 ppt salinity [13]. Studies have shown that, compared with the freshwater stress group (0 psu), the expression of migration inhibitory factor-2 (MIF-2), transforming growth factor-β1 (TGF-β1), tumor necrosis factor α (TNF-α), and interleukin-1 (IL-1) genes are significantly higher in the salinity stress group (16 psu) in the spleen of Nile tilapia [14]. Salinity affects the expression of immune genes in fish, but from the perspective of the HPI axis, the effect of salinity on the expression of immune genes has rarely been analyzed.

Large yellow croaker (*Larimichthys crocea*) belongs to the order Perciformes, family Sciaenidae, and genus *Larimichthys* [15]. It is a euryhaline species, meaning it can tolerate a range of salinities, but optimal growth and health are highly dependent on specific salinity conditions. However, the artificial breeding technology of large yellow croaker faces various problems and challenges, such as susceptibility to disease and a low survival rate [16]. In the natural environment where large yellow croaker lives, it is often affected by exogenous factors such as temperature, salinity, starvation, and heavy metal pollution. As the salinity of seawater will change due to the influence of tides, large yellow croaker will experience salinity stress for more than several months during this period, which will have a certain degree of impact on the physiological functions of fish growth, energy metabolism, and immunity [17]. Therefore, understanding how salinity influences immune-related gene expression is crucial for improving aquaculture practices and ensuring sustainable production of large yellow croaker. This study explored the expression changes of the HPI axis and immune-related genes in various tissues of large yellow croaker under salinity stress and analyzed the regulatory effect of the HPI axis genes on the expression of downstream immune factors. The results will help to further understand the HPI axis and immune-related molecular regulatory mechanisms of large yellow croaker under environmental stress and provide a reference for the selection of seawater salinity in aquaculture in the future.

## 2. Results

### 2.1. Sequence Analysis and De Novo Assembly for Large Yellow Croaker

All datasets from the DNA Nanoball Sequencing (DNBSEQ) platform can be found in the Short Read Archive (SRA) database of the National Center for Biotechnology Information (NCBI) under accession number SRP315057. Averages of 130.18 M, 129.52 M, and 130.15 M clean reads were generated by filtering the raw reads from normal salinity (NB), low salinity (LB), and high salinity (HB), respectively. The sequencing quality of the three sets of transcriptome data was tested (each set was repeated 3 times), and it was found that Q20 was above 96%, Q30 was above 93%, and the Clean Reads Ratio was above 93% (Table 1).

### 2.2. Differentially Expressed Gene Analysis

This experiment analyzed the changes in differentially expressed genes in the transcriptome of large yellow croaker brains under different salinity stresses. The results showed that there were 1568 differentially expressed genes in the HB and NB groups, of which 494 were significantly up-regulated and 1074 were significantly down-regulated. There are 1720 differentially expressed genes in the LB and NB groups, of which 486 are significantly up-regulated and 1234 are significantly down-regulated (Figure 1). In the HB and NB groups, differentially expressed genes (DEGs) mainly involve some pathways related to transportation, catabolism, and immune regulation, such as cardiac muscle contraction, long-term potentiation, the oxytocin signaling pathway, oxidative phosphorylation, thermogenesis, retrograde endocannabinoid signaling, axon guidance, adrenergic signaling in cardiomyocytes, the mitogen-activated protein kinase (MAPK) signaling pathway, the cyclic guanosine monophosphate-protein kinase G (cGMP-PKG) signaling pathway, and the calcium signaling pathway (Figure 2). In the LB and NB groups, DEGs are mainly related to neural and immune regulation, such as insulin secretion, glutamatergic synapse, aldosterone synthesis and secretion, the oxytocin signaling pathway, axon guidance, circadian entrainment, long-term potentiation, the cyclic adenosine monophosphate (cAMP) signaling pathway, proximal tubule bicarbonate reclamation, and adrenergic signaling in cardiomyocytes.

### 2.3. mRNA Expression Levels of HPI Axis and Immune-Related Genes

The quantitative Real-time polymerase chain reaction (qPCR) results showed that the expression level of *β2-ADR* in the kidney was significantly up-regulated under salinity conditions of 6, 12, 30, and 36 ppt (*p* < 0.05), while there was no significant difference at a salinity of 18 ppt. It is up-regulated in the muscle (except 6W) (*p* < 0.05). In the spleen, *β2-ADR* was significantly up-regulated at salinity values of 6 and 12 ppt (*p* < 0.05) (Figure 3). *GH* expression is highest in the brain and gill, with *GH* significantly up-regulated in the gill under salinity conditions of 6, 12, and 18 ppt (*p* < 0.05). *GH* expression was also up-regulated in the liver, spleen, and other tissues, but there was no significant difference in expression in the intestine (Figure 4). *PRL* was only significantly expressed in the brain and consistently up-regulated from 4W to 8W (*p* < 0.05) (Figure 5).

To explore whether the HPI axis can regulate the expression of immune-related genes, we selected two immune pathways that significantly enriched in transcriptome data, namely, the cytokine–cytokine receptor interaction and prolactin signaling pathway, and selected eight immune-related genes, interleukin-6 signal transducer (*IL6st*), interleukin-6 (*IL6*), cluster of differentiation 40 (*CD40*), CXC motif chemokine ligand 12 (*CXCL12*), interferon alpha and beta receptor subunit 1 (*IFNAR1*), interferon regulatory factor 1 (*IRF1*), suppressors of cytokine signaling 2 (*SOCS2*), and suppressors of cytokine signaling 6 (*SOCS6*) for expression analysis. It was found that these genes expression have significant spatiotemporal specificity. Under salinity stress, most genes (such as *IL6st*, *CD40*, and *CXCL12*) show significant up-regulation in liver, muscle, gill, and other tissues, and their expression patterns dynamically change with stress duration. *IL6st* is highly expressed in the liver and muscle at 2W, transferred to the gill at 6W, and restored to the muscle and intestine at 8W (Figure 6). The expression of *IL6* was inhibited at low salinity and activated at high salinity. At 6W, there was a significant up-regulation in the spleen, liver, and kidney (Figure 7). *CD40* exhibits salinity response in muscles at 2W, is up-regulated in the kidney under low salinity stress at 4W, and is down-regulated under high salinity stress. At 8W, it is highly expressed in the intestine and spleen (Figure 8). *CXCL12* is up-regulated in the intestine, liver, and muscle from 2W to 4W, with the salinity response in the gill at 6W and a peak in the intestine and spleen at 8W (Figure 9). *IFNAR1* is expressed in the spleen, liver, kidney, and muscle from 2W to 4W, while muscles have a high salinity response. Highly expressed in the gill between 6W and 8W (Figure 10), *IRF1* exhibits temporal tissue transfer and overall up-regulation of expression levels (Figure 11). *SOCS2* exhibits a multiple-tissue salinity response (Figure 12). *SOCS6* is up-regulated with salinity in the muscle and gill. The expression was activated at low salinity in the kidney and inhibited at high salinity (Figure 13). Correlation analysis shows that there is a significant regulatory association between immune genes and HPI axis key genes (*β2-ADR*, *GH*). In addition, salinity stress affects regulatory intensity (such as the significant correlation between *CXCL12* and *β2-ADR* under low salinity conditions). The results suggested that the HPI axis dynamically regulates immune gene expression through neuroendocrine factors such as *GH*, and the duration and intensity of salinity stress are important regulatory factors.

## 3. Discussion

*β2-ADR* plays an important role in regulating a variety of biological functions, such as participating in the synthesis of biological materials required for post-traumatic tissue repair [18,19], which has been studied in mammals more maturely. However, the research on *β2-ADR* in fish is still very limited, and the function distribution in large yellow croaker is even more unknown. In this study, we found that *β2-ADR* has the highest expression in the kidney and muscle, followed by the liver and spleen, and lower expression in the spleen, gill, and brain. Regarding the high expression of *β2-ADR* in the muscle of large yellow croaker, we speculate that this may be related to the maintenance of the muscle stability of large yellow croaker under salinity stress. Studies have shown that *β2-ADR* is widely present in smooth muscle and skeletal muscle and plays a role in maintaining the stability of muscle structure [20], and the interference of exogenous *β2-ADR* agonists can also cause the symptoms of muscle hypertrophy [21]. These results can be used as a basis for the higher expression level of *β2-ADR* in the muscle. In addition, *β2-ADR* also has a higher expression level in kidney tissue, and its expression level is greatly increased with the change in salinity. This indicates that *β2-ADR* may participate in endocrine regulation under salinity stress through the HPI axis of large yellow croaker.

Moreover, we found that *GH* is expressed in various tissues of large yellow croaker, and the expression level in the brain and gill is relatively high. Some studies have found that *GH* is mainly expressed in the pituitary, spleen, heart, and gill tissues in adult *Epinephelus coioides* [22]. Another study found that the expression of *GH* in the liver, spleen, brain, gonads, and pituitary increases sequentially in *Epinephelus lanceolatus* [23]. Factors secreted by the hypothalamus and surrounding tissues are responsible for the production of *GH* by pituitary cells. In bony fish, pituitary adenylate cyclase-activating peptide (*PACAP*) and corticotropin-releasing hormone (*CRH*) can promote *GH* synthesis and secretion and activate cascade signal transduction pathways [24]. This indicates that the expression of *GH* in fish brains may be related to a series of hormonal cascades downstream of the hypothalamus mediated by the HPI axis. In addition, the expression of *GH* in the gill is primarily associated with the regulation of osmotic pressure in large yellow croaker under salinity stress [25].

*PRL* is a hormone with multiple functions in regulating biological homeostasis. We found that *PRL* is mainly expressed in the brain of large yellow croaker, while its expression is lower in other tissues. Studies have found that, under low salinity conditions, the *PRL* gene of *Scatophagus argus* is only expressed in brain and pituitary tissues, and the abundance is extremely high, which is consistent with our results [26]. In our study, the expression of *PRL* has certain tissue limitations in large yellow croaker, which may be due to the fact that *PRL* does not occupy a very important position as an osmotic pressure regulator but has other physiological functions. For example, the osmotic pressure adjustment of salmon in the process of adapting to a freshwater environment is not closely related to *PRL* [27,28]. In addition, *PRL* can be used as a mediator in the immune–neuroendocrine regulatory network. It can bind to *PRL* by means of trimers and endow it with biological activity and transmit information into the nucleus through the signal transduction of Janus kinase 2 (JAK2) and signal transducer and activator of transcription (STAT) to cause the target gene transcription and expression [7]. *PRL* can also promote the proliferation of immune cells such as T lymphocytes and Nb2 lymphoma cell lines [29]. Therefore, *PRL* may be mainly involved in immune–neuroendocrine regulation under the salinity stress of large yellow croaker, while osmotic pressure regulation mainly depends on *GH*.

*IL-6* is a pleiotropic cytokine secreted by a variety of cells and participates in immune and inflammatory responses at inflammatory sites [30]. *IL-6* exerts its extensive biological activities by interacting with *IL6st* [31]. However, the current understanding of the function of *IL6st* in bony fish is limited, and only grass carp have shown that *IL6st* can mediate the immunomodulatory function of *IL-6* [32]. Studies have confirmed that *IL-6* plays a role in molecular signaling in the interaction between the immune system and the HPI axis. In our study, the expression trends of *IL6* and *IL6st* are basically the same, indicating that salinity stress activates the *IL6st*-mediated *IL6* signal transduction mechanism in large yellow croaker, and this transmission mechanism may be regulated by the HPI axis [33]. Secondly, salinity stress also caused an increase in the expression of *CXCL12* in various immune tissues of large yellow croaker. It has been reported that significantly higher expression levels have been detected in immune-related organs, including the head kidney, spleen, and kidney in rock seabream (*Oplegnathus fasciatus*) [34]. The combination of *CXCL12* and its receptor of CXC motif chemokine receptor 4 (*CXCR4*) activates the G protein-coupled receptor signaling pathway and regulates a series of physiological behaviors mediated by environmental stress, neuropeptides, hormones, growth factors, nucleotides, etc. [35,36]. Our results showed that the expression of the *CXCL12* gene in the immune tissue of large yellow croaker under salinity stress was down-regulated, which may be negatively regulated by neuroendocrine-related genes. Our results also indicate that salinity stress may trigger the inflammatory mechanism of large yellow croaker and that key signal transduction pathways in the neuroendocrine–immune pathway are involved in regulation. *CD40* has been confirmed in Japanese flounder as an accessory molecule of the major histocompatibility complex with cluster of differentiation 69 (*CD69*) to directly participate in the immune response process against *Vibrio anguillarum* infection [37]. As a prerequisite for the activation of the Janus kinase-signal transducer and activator of transcription (JAK-STAT) signaling pathway, *IFNAR* has been confirmed in several bony fishes [38,39,40]. The expression level of *IFNAR* plays an important role in regulating the expression of antiviral genes [41]. In this study, *CD40* and *IFNAR1* were up-regulated in the immune tissues of large yellow croaker, indicating that salinity stress may affect the anti-pathogen ability, reduce immunity, and adversely affect the resistance of large yellow croaker to foreign virus invasion. In addition, our results showed that the expression of these immune factors in the intestine was significantly increased when the salinity stress progressed to the later stage. This may be due to the long-term salinity stress causing inflammation to spread from other tissues of the large yellow croaker to the intestine.

The three regulatory factors *SOCS2*, *SOCS6*, and *IRF1* were expressed in various tissues of large yellow croaker in this study. Among them, the expression of the intestine, muscle, and gill was higher, followed by immune tissues, such as spleen and kidney tissues. Another study found that the expression of *IRF1* was highest in the gill and spleen of large yellow croaker but lower in the brain [42]. In Japanese flounder, *IRF1* is also mainly expressed in the intestine, muscle, liver, heart, and spleen, while the expression is relatively low in the brain and kidney [43]. These results are basically consistent with our results. In addition, when Japanese flounder were infected with an exogenous virus, the expression of *IRF1* increased several times compared to normal. In our study, *IRF1* also has different degrees of up-regulation under different salinities, which also implies that salinity stress may affect the virus susceptibility of large yellow croaker. Our results show that *SOCS2* was highly expressed in the liver, kidney, and intestine of large yellow croaker, followed by the gill and brain. This is consistent with the results of the high expression of *SOCS2* in the brain, head kidney, muscle, spleen, gill, skin, and intestine in rainbow trout [44]. The up-regulation of *SOCS2* was significantly higher than other salinities under the condition of a salinity of 36 ppt, indicating that high salinity has a more significant effect on the expression of *SOCS2* compared with low salinity. In addition, the high expression of *SOCS6* in the kidney, gill, and brain of large yellow croaker is also consistent with the tissue expression specificity of *SOCS6* in Japanese flounder and rainbow trout [45,46]. Studies have shown that the expression of *SOCS* is regulated by cortisol signals, while cortisol, as a sign of HPI axis activation, is regulated by the HPI axis and participates in the response to environmental stress [47].

The correlation analysis of the expression levels of immune-related genes and HPI axis-related genes revealed that the expression of these immune genes seems to have a certain regulatory relationship with *β2-ADR*, *GH*, and *PRL*. In other words, salinity stress causes HPI axis activation and mediates the occurrence of an immune response. There is evidence to support the anti-inflammatory role of *β2-ADR* in inflammation [48]. In vitro adrenaline stimulation can inhibit respiratory bursts and the expression levels of pro-inflammatory cytokines *TNF-α* and interleukin-12 (*IL-12*) [49,50,51], and stimulate the secretion of anti-inflammatory cytokine interleukin-10 (*IL-10*) [52]. In addition, *β2-ADR* can increase *IL-6* and *TNF-α* in neonatal rat cardiac fibroblasts by mediating the non-classical cAMP pathway and the p38 mitogen activated protein kinases (p38-MAPK) pathway [19]. In our study, *IL6* expression was up-regulated in the liver, kidney, and other immune tissues affected by salinity, and *IL6* and *β2-ADR* expression changes are positively correlated, which may be because *β2-ADR* participates in the regulation of *IL6* under salinity stress anti-inflammatory responses, and similar expression changes of *IL6st* may be indirectly regulated by *IL6*. In addition, the effect of the *β2-ADR*-derived signal on nuclear factor-kappaB (NF-κB) activity in immune cells seems to have a high degree of cell-type specificity and gene selectivity [48]. *CD40* can mediate the activation of the NF-κB signaling pathway to promote the activation of B lymphocytes [53]. Adrenaline can also inhibit the expression of pro-inflammatory factors, chemokines, and their receptors [54]. Our results found that the expression of *CD40* and *CXCL12* was down-regulated in the intestine, spleen, kidney, and other tissues and also reflected the inhibitory effect of *β2-ADR* on cytokines at the transcription level.

Both *GH* and *PRL* participate in immune regulation, and a large number of signal molecules have since expanded and activated similar intracellular signal transduction cascades, such as signal transducers of the JAK family and STAT family. Studies have shown that LPS interacts with Toll-like receptors (*TLRs*) and increases growth hormone releasing hormone (*GHRH*) to regulate the release of *GH* [55]. On the other hand, *TNF-α*, *IL-1β*, and *IL-6* down-regulate the expression of growth hormone receptor (*GHR*) in the liver and indirectly negatively regulate *GH* [56]. In our study, the expression of *GH* and *IL6* showed a negative correlation in the liver under salinity conditions of 18 and 30 ppt, but other tissues did not. This shows that under, these conditions, the liver may transmit signals through the regulation mechanism of *IL6* and *GH* to cope with salinity pressure. Other organizations may have different mechanisms to respond to external stimuli because of their different functions. Studies have shown that, in the thymus and head kidney of transgenic zebrafish overexpressing *GH*, the expression of recombination-activating gene 1 (*RAG-1*), interleukin-1α (IL-1α), and cluster of differentiation 4 (*CD4*) genes has been reduced [57,58], indicating that excessive *GH* impairs their immune function. In addition, the signal transduction of the *GH* signaling pathway was regulated by *SOCS*. In human muscle cells, *GH* can increase the expression of *SOCS1*, *SOCS2*, *SOCS3*, and cytokine-inducible SH2-containing protein (*CIS*) [59]. Our results showed that the down-regulation of *SOCS2* and the up-regulation of *GH* under low salinity conditions directly reflected the negative regulatory relationship between the two. However, the up-regulation of *SOCS2* and the down-regulation of *GH* under high salinity conditions also reveal that the up-regulation of *SOCS2* may also be related to the significant up-regulation of *PRL* during the same period. In addition, *SOCS2* also has a similar dual regulatory effect on *PRL* signal transduction [60]. Studies have found that the physiological concentration of natural *PRL* can induce the expression of pro-inflammatory cytokines *IL-1β* and *TNF-α* and the production of reactive oxygen species in the head kidney white blood cells and macrophages of the gilthead seabream, and it can pass the JAK/STAT and NF-κB signaling pathway, which promotes the pro-inflammatory classic M1 polarization of snapper macrophages [61]. Similar results also showed the production of superoxide anions, phagocytic activity, and lysozyme levels induced by *PRL* [62]. In summary, the neuroendocrine–immune pathway of large yellow croaker is fully activated under salinity stress.

## 4. Materials and Methods

### 4.1. Treatment of Fish Salinity Stress

A total of 1000 large yellow croakers with an average weight of (40 ± 5.42) g were provided by Ningbo Xiangshan Harbor Aquatic Seed Co., Ltd., in Ningbo, China, and were temporarily reared in Ningbo Ocean and Fishery Technology Innovation Base. Before the experimental treatment, all the experimental fish were domesticated for one week in normal seawater at a water temperature of about 23 °C (normally fed and protected from light). One week later, the domesticated large yellow croaker were randomly divided into 6 experimental groups with different salinity (low salinity of 6, 12, and 18 ppt, normal salinity of 24 ppt, and high salinity of 30 and 36 ppt). Each group had three parallel barrels, and each barrel (the specification is outer diameter × bottom diameter × height = 1.07 m × 0.88 m × 0.85 m) had 50 large yellow croakers. The aquaculture temperature was 23 °C, the dissolved oxygen was 6.5 mg/L, and the pH was 7.8–8.0. All groups were fed normally during the aquaculture period. At first, 10 individual large yellow croaker tissues, including gonad, intestine, spleen, stomach, liver, kidney, heart, muscle, gill, and brain tissues, were randomly selected at 0 weeks as control samples. Then, 10 individual large yellow croaker tissues were randomly selected from different salinity groups in 2, 4, 6, and 8 weeks, respectively. The collected tissue was placed in a 2.0 mL RNase-free cryotube and temporarily placed in a liquid nitrogen environment and then stored in a −80 °C refrigerator for a long time for RNA extraction.

### 4.2. RNA Extraction and Quantification

Trizol (Thermo Fisher Scientific, Waltham, MA, USA) reagent was used to extract total RNA from three groups of large yellow croaker brain tissues (salinity of 12, 24, and 36 ppt) at the 4W period. The quality of RNA was monitored on a 1% agarose gel. A NanoDropND-2000 ultra-micro spectrophotometer (Thermo Fisher Scientific, Waltham, USA) was used for RNA concentration detection. The qualified total RNA was used for library preparation, clustering, and classification of transcriptome sequencing as well as transcriptome library preparation and sequencing.

Three group samples of large yellow croaker brain tissue from the low salinity group (12 ppt), the normal group (24 ppt), and the high salinity group (36 ppt) were used for transcriptome sequencing and named low salinity (LB), normal salinity (NB), and high salinity (HB), respectively. Each group of samples had 3 independent individuals. The entire libraries of three group samples were constructed by using the Mate Pair Library Preparation Kit (BGI Genomics, Shenzhen, China) according to the manufacturer’s instructions, respectively. Finally, these libraries were sequenced using DNA Nanoball Sequencing (DNBSEQ) (BGI Genomics, Shenzhen, China).

The assembly of transcripts, annotation, and the identification of the differentially expressed genes were conducted. We filtered the raw data obtained by DNBSEQ sequencing to remove impurities in the raw reads. The filtered clean reads were aligned to the reference genome using hierarchical indexing for spliced alignment of transcripts (HISAT) to obtain the position information and alignment number of each sequence in the gene, and then we performed a new transcript prediction, single nucleotide polymorphism (SNP) and insertion and deletion (InDel), and differentially spliced gene detection. Then, we used Bowtie to align the clean reads to the genome sequence, and then we used RNA-Seq by Expectation-Maximization (RSEM) to calculate the gene expression level of each sample [63,64]. Finally, for multiple samples, we detected the differentially expressed genes (Q value (Adjusted *p* value) ≤ 0.05 as differentially expressed genes) between different samples according to requirements [65,66], and we performed Gene Ontology (GO) function enrichment analysis and Kyoto Encyclopedia of Genes and Genomes (KEGG) enrichment analysis on differentially expressed genes. The hypergeometric distribution test of the R software, version 4.3.2 package was used for enrichment analysis, and the function with Q value ≤ 0.05 was regarded as significant enrichment.

### 4.3. Quantitative Real-Time PCR Analysis

Total RNA was extracted from intestine, spleen, liver, kidney, muscle, gill, and brain tissues in different salinity groups (low salinity of 6, 12, and 18 ppt, normal salinity of 24 ppt, and high salinity of 30 and 36 ppt) using the trizol method for 0W, 2W, 4W, 6W, and 8W. The quality of RNA was monitored on a 1% agarose gel, and the concentration of RNA was detected using a NanoDropND-2000 ultra-micro spectrophotometer (Thermo Fisher Scientific, Waltham, USA). ReverTra Ace qPCR RT Master Mix with a gDNA Remover kit (Toyobo, Osaka, Japan) was used for cDNA synthesis, and Primer software, version 5.0 was used for primer design. The qPCR was performed using THUNDERBIRD SYBR^®^ qPCR Mix (Toyobo, Osaka, Japan) on ABI 7500 Fast (Applied Biosystems, Foster, USA). The qPCR reaction volume contained 2 μL cDNA, 0.8 μL each primer, 0.4 μL ROX Reference DyeII, 10 μL SYBR Green IMaster (Takara, Kanazawa, Japan), and 6 μL water. PCR amplification was performed under the following conditions: 95 °C for 30 s, followed by 40 cycles of 95 °C for 3 s and 60 °C for 30 s. All samples were measured in triplicate. Normalization and fold changes were calculated using the 2^−ΔΔCT^ method. Differences in HPI axis and immune-related gene expression were evaluated by unpaired t-test. All analyses and graphs were performed using GraphPad Prism software, version 6.0.

## 5. Conclusions

This study reveals the key mechanism by which salinity stress affects the neuroendocrine–immune regulatory network of large yellow croaker. By integrating transcriptomics and molecular validation techniques, we found that HPI axis-related genes (*β2-ADR*, *GH*, and *PRL*) and immune genes (*CXCL12* and *SOCS2*) exhibit spatiotemporal specific co-expression characteristics during salinity adaptation, and their interaction patterns dynamically change with stress intensity and duration. These results provide a new perspective for analyzing the salinity adaptation mechanism of fish. The HPI axis and immune interaction regulation model established in this study can serve as a theoretical basis for salinity regulation strategies in aquaculture. By optimizing salinity parameters, improved variety selection and precise environmental management can be achieved, which has practical value for enhancing the survival rate and disease resistance of large yellow croaker aquaculture.

## Figures and Tables

**Figure 1 ijms-26-04298-f001:**
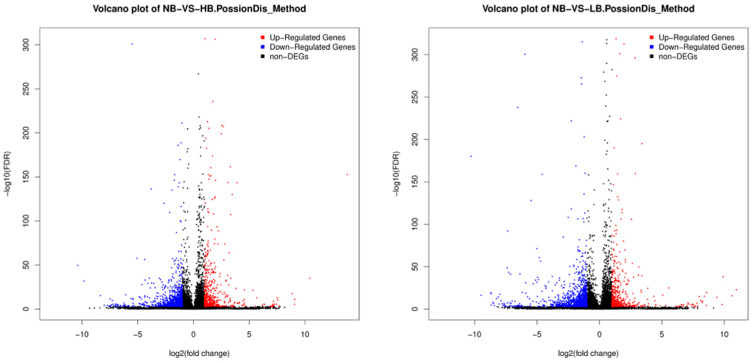
Volcano map of differentially expressed genes between HB vs. NB and LB vs. NB.

**Figure 2 ijms-26-04298-f002:**
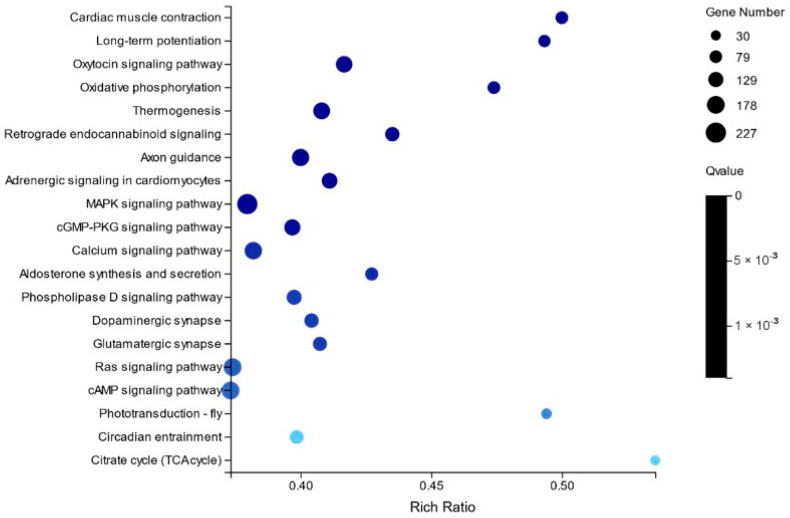
NB vs. HB scatter plot of KEGG enrichment of differential genes. The X-axis is the enrichment ratio. The Y-axis is the KEGG pathway, the size of the bubble represents the number of genes annotated to a KEGG pathway, and the color represents the enriched Q value, and the darker the color, the smaller the Q value.

**Figure 3 ijms-26-04298-f003:**
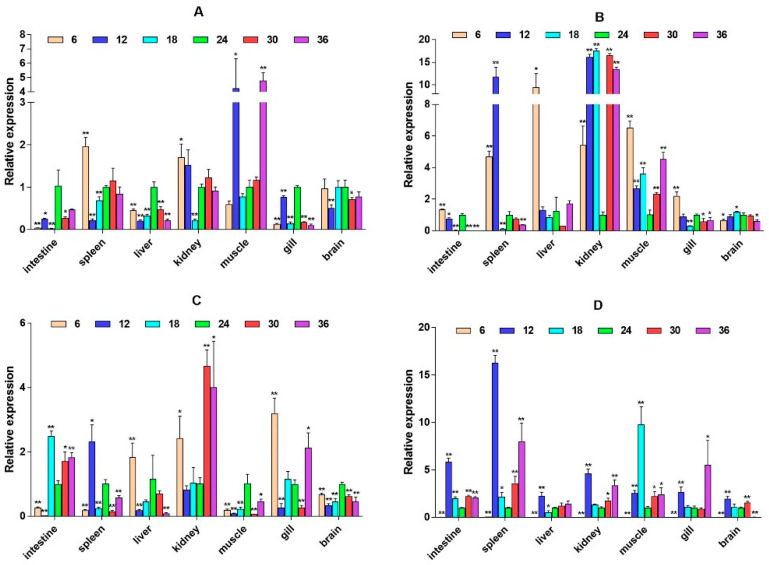
Relative mRNA expression level of the *β2-ADR* gene. (**A**–**D**) are, respectively, 2W, 4W, 6W, and 8W, i.e., four periods; the square on the above represents salinity. * *p* < 0.05 indicates significant difference, and ** *p* < 0.01 indicates a highly significant difference.

**Figure 4 ijms-26-04298-f004:**
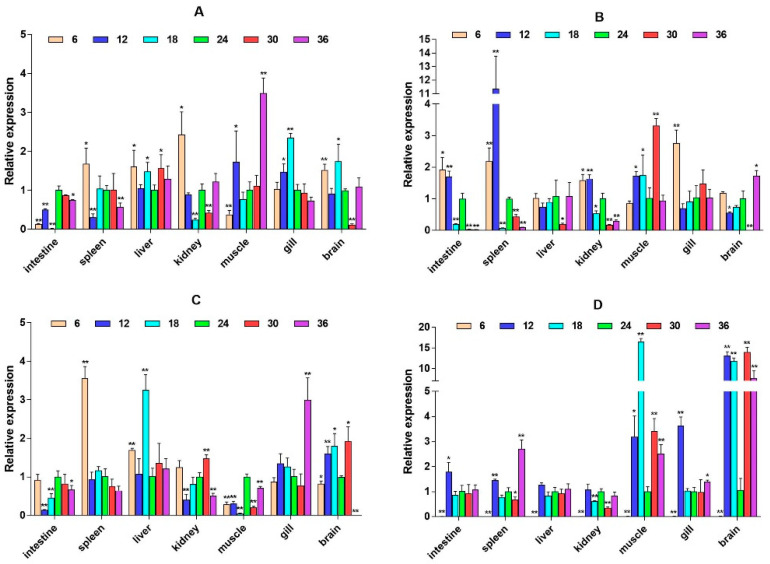
Relative mRNA expression level of the *GH* gene. (**A**–**D**) are, respectively, 2W, 4W, 6W, and 8W, i.e., four periods; the square on the above represents salinity. * *p* < 0.05 indicates significant difference, and ** *p* < 0.01 indicates a highly significant difference.

**Figure 5 ijms-26-04298-f005:**
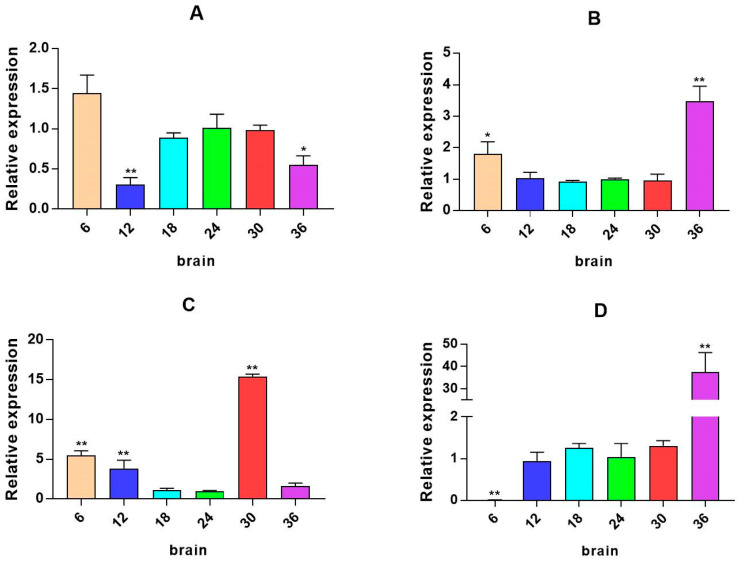
Relative mRNA expression level of the *PRL* gene. (**A**–**D**) are, respectively, 2W, 4W, 6W, and 8W, i.e., four periods; different colored bar charts represents salinity mRNA expression levels of immune-related genes. * *p* < 0.05 indicates significant difference, and ** *p* < 0.01 indicates a highly significant difference.

**Figure 6 ijms-26-04298-f006:**
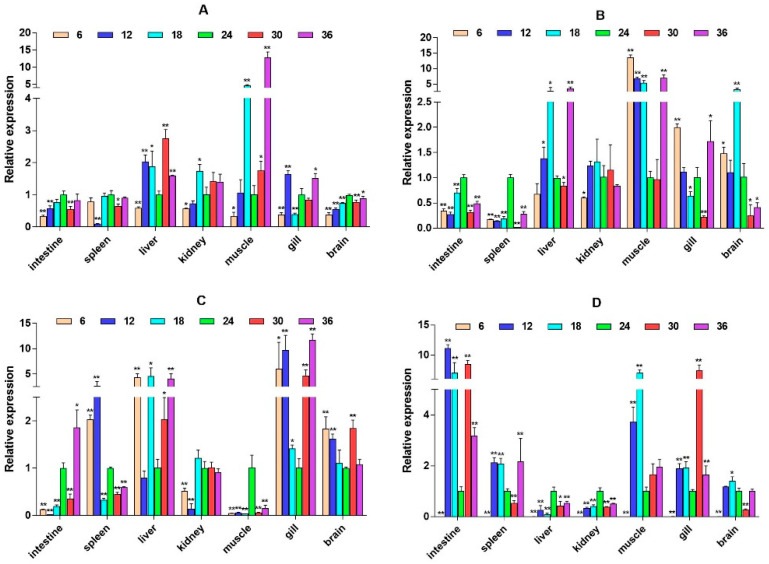
Relative mRNA expression level of the *IL6st* gene. (**A**–**D**) are, respectively, 2W, 4W, 6W, and 8W, i.e., four periods; the square on the above represents salinity. * *p* < 0.05 indicates significant difference, and ** *p* < 0.01 indicates a highly significant difference.

**Figure 7 ijms-26-04298-f007:**
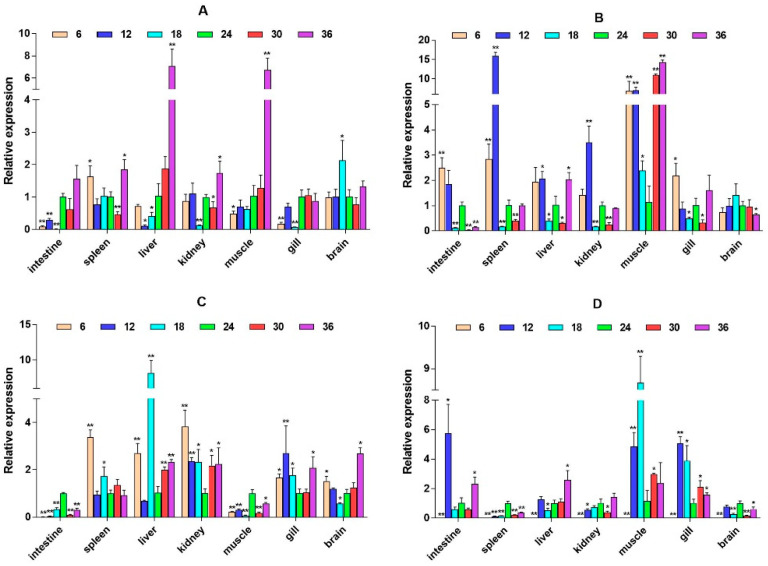
Relative mRNA expression level of the *IL6* gene. (**A**–**D**) are, respectively, 2W, 4W, 6W, and 8W, i.e., four periods; the square on the above represents salinity. * *p* < 0.05 indicates significant difference, and ** *p* < 0.01 indicates a highly significant difference.

**Figure 8 ijms-26-04298-f008:**
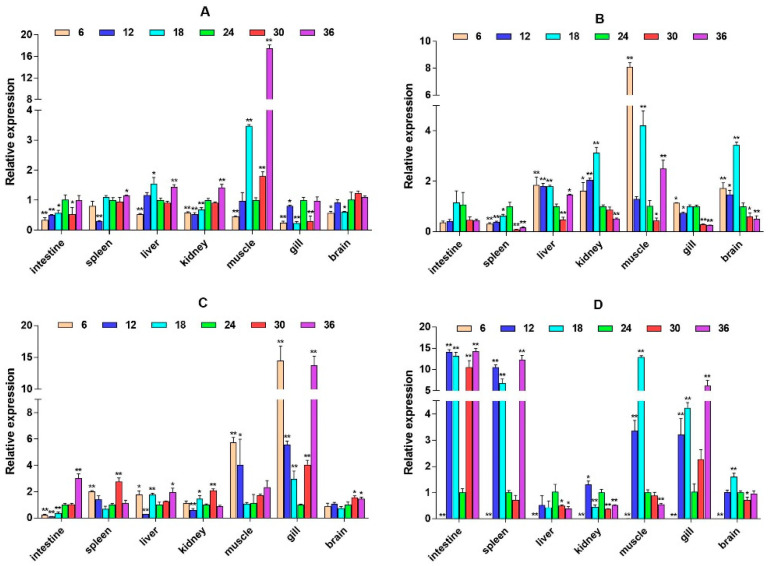
Relative mRNA expression level of the *CD40* gene. (**A**–**D**) are, respectively, 2W, 4W, 6W, and 8W, i.e., four periods; the square on the above represents salinity. * *p* < 0.05 indicates significant difference, and ** *p* < 0.01 indicates a highly significant difference.

**Figure 9 ijms-26-04298-f009:**
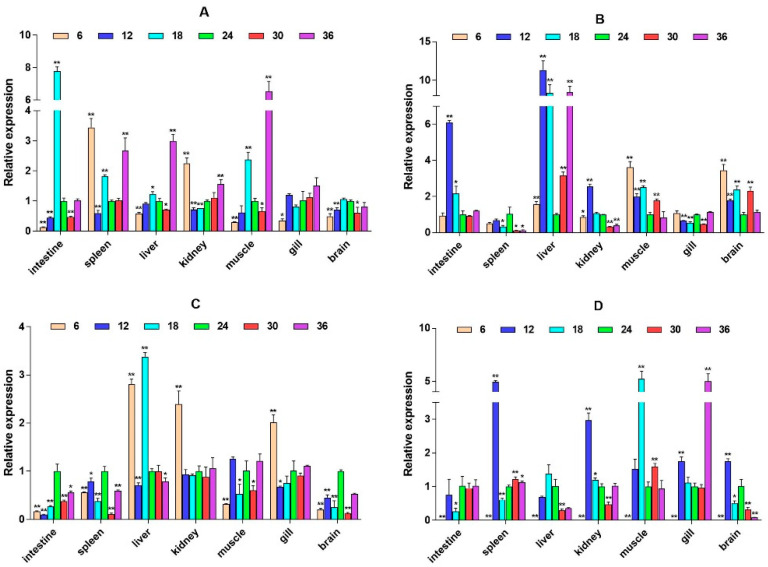
Relative mRNA expression level of the *CXCL12* gene. (**A**–**D**) are, respectively, 2W, 4W, 6W, and 8W, i.e., four periods; the square on the above represents salinity. * *p* < 0.05 indicates significant difference, and ** *p* < 0.01 indicates a highly significant difference.

**Figure 10 ijms-26-04298-f010:**
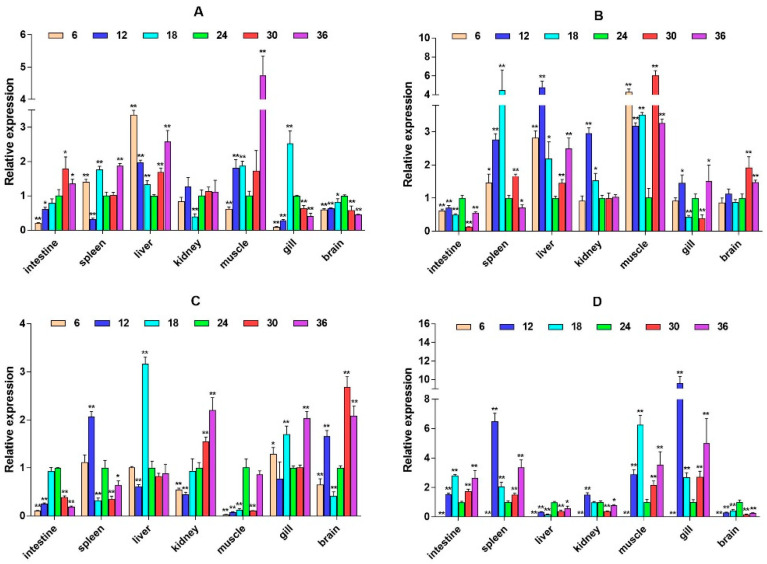
Relative mRNA expression level of the *IFNAR1* gene. (**A**–**D**) are, respectively, 2W, 4W, 6W, and 8W, i.e., four periods; the square on the above represents salinity. * *p* < 0.05 indicates significant difference, and ** *p* < 0.01 indicates a highly significant difference.

**Figure 11 ijms-26-04298-f011:**
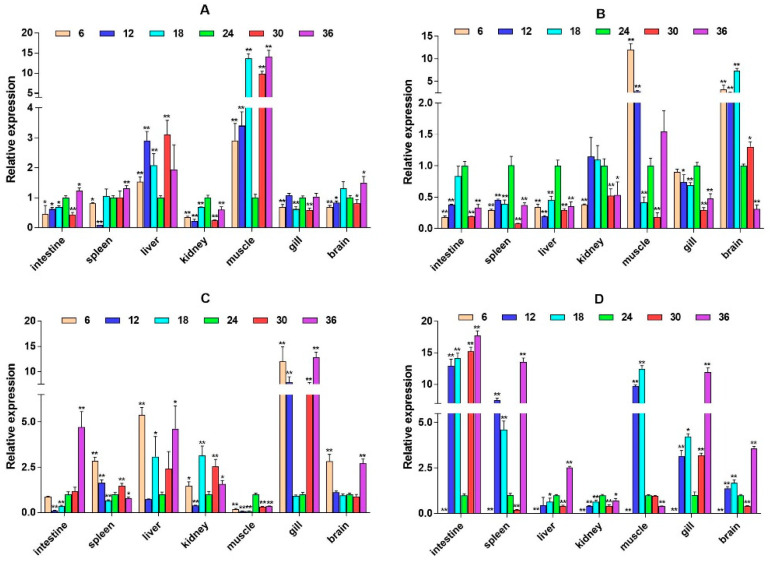
Relative mRNA expression level of the *IRF1* gene. (**A**–**D**) are, respectively, 2W, 4W, 6W, and 8W, i.e., four periods; the square on the above represents salinity. * *p* < 0.05 indicates significant difference, and ** *p* < 0.01 indicates a highly significant difference.

**Figure 12 ijms-26-04298-f012:**
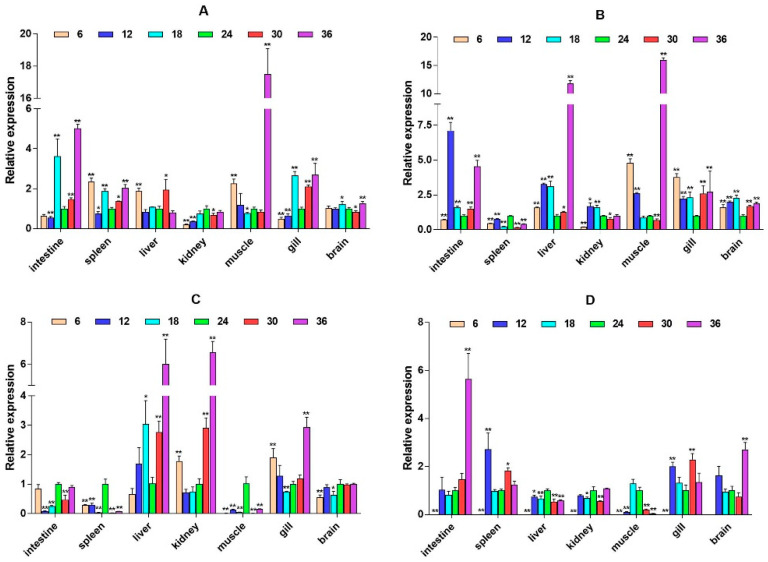
Relative mRNA expression level of the *SOCS2* gene. (**A**–**D**) are, respectively, 2W, 4W, 6W, and 8W, i.e., four periods; the square on the above represents salinity. * *p* < 0.05 indicates significant difference, and ** *p* < 0.01 indicates a highly significant difference.

**Figure 13 ijms-26-04298-f013:**
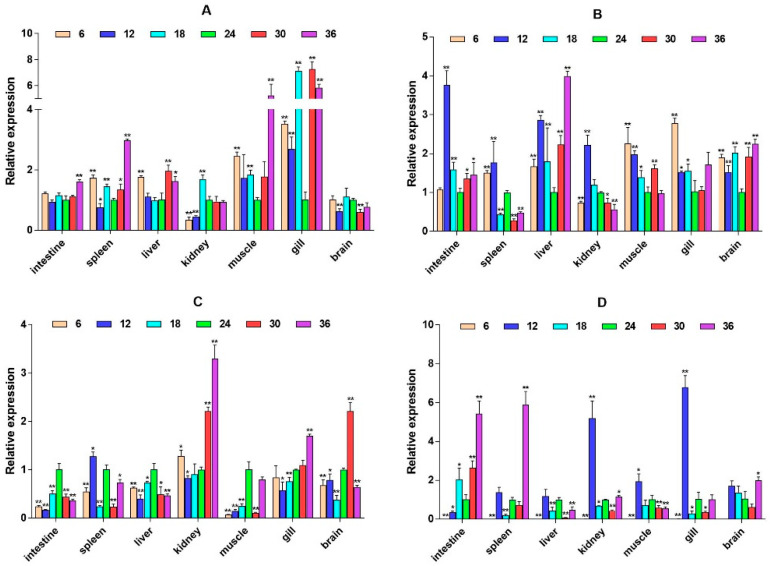
Relative mRNA expression level of the *SOCS6* gene. (**A**–**D**) are, respectively, 2W, 4W, 6W, and 8W, i.e., four periods; the square on the above represents salinity. * *p* < 0.05 indicates significant difference, and ** *p* < 0.01 indicates a highly significant difference.

**Table 1 ijms-26-04298-t001:** The output statistics of transcriptome sequencing.

Sample	Total Raw Reads (M)	Total Clean Reads (M)	Total Clean Bases (Gb)	Clean Reads Q20 (%)	Clean Reads Q30 (%)	Clean Reads Ratio (%)
HB1	43.82	43.37	6.51	98.11	94.62	98.97
HB2	43.82	43.4	6.51	97.88	93.97	99.05
HB3	43.82	43.38	6.51	97.96	94.18	99
LB1	45.57	42.69	6.4	96.82	88.07	93.66
LB2	43.82	43.41	6.51	97.83	93.8	99.07
LB3	43.82	43.42	6.51	97.87	93.93	99.08
NB1	43.82	43.38	6.51	98	94.31	98.99
NB2	43.82	43.43	6.51	97.97	94.19	99.11
NB3	43.82	43.37	6.51	98	94.32	98.97

Q20 and Q30 are the main indicators for weighing the reliability of transcriptome sequencing data, indicating the percentage of the total clean reads with a quality value greater than 20 bp.

## Data Availability

The original contributions presented in this study are included in the article. Further inquiries can be directed to the corresponding authors.

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
