# Peer review of "Regulation of Immune-Related Gene Expression by Salinity-Induced HPI Axis in Large Yellow Croaker, Larimichthys crocea"

_ijms, 2025, doi:10.3390/ijms26094298_

Round 1

Reviewer 1 Report

Comments and Suggestions for Authors

In the present study, cultured large yellow croaker was used as the research object, and RNA-seq technology was used to sequence the transcription group of different salinity (12, 24, 36) treatment of large yellow croaker brain tissue for 4 weeks. The results proved that salinity stress can activate the HPI axis of large yellow croaker, and can also affect the immune function of large yellow croaker, and the expression of immune factors in the immune response is regulated by the upstream gene in the HPI axis. In general, this is a interesting topic.

1. Line 24-24, In addition, The Western Blot results also showed that the protein expression of CXCL12 and SOCS2 was tissue-specific and affected by salinity stress. The Western Blot should be the Western Blot.

2. Line 25-28, In summary, this experiment proves that salinity stress can activate the HPI axis of large yellow croaker, and can also affect the immune function of large yellow croaker, and the expression of immune factors in the immune response is regulated by the upstream gene in the HPI axis.. There is an extra period at the end of the sentence.

3. Line 90, What are NB, LB and HB?

4. Line 158-160, In kidney tissue, β2-ADR was significantly up-regulated under the conditions of salinity 6, 12, 30, and 36, but there was no significant difference under the salinity of 18. why?

5. Line 261-262, Western Blot verified the protein expression level of the two genes CXCL12 and SOCS2. Why did you choose only these two proteins?

Comments on the Quality of English Language

 The English could be improved to more clearly express the research.

Author Response

Reviewer #1:

In the present study, cultured large yellow croaker was used as the research object, and RNA-seq technology was used to sequence the transcription group of different salinity (12, 24, 36) treatment of large yellow croaker brain tissue for 4 weeks. The results proved that salinity stress can activate the HPI axis of large yellow croaker, and can also affect the immune function of large yellow croaker, and the expression of immune factors in the immune response is regulated by the upstream gene in the HPI axis. In general, this is a interesting topic.

Reply: Thank you for your constructive comments on our manuscript. These questions are very helpful in improving the quality and clarity of our manuscript. We have made detailed revisions to the manuscript and checked every detail in it. The response to each of your comments are as follows. Please criticize and correct.

1. Line 24-24, In addition, The Western Blot results also showed that the protein expression of CXCL12 and SOCS2 was tissue-specific and affected by salinity stress. The Western Blot should be the Western Blot.

Reply: Thanks for your valuable comments on our manuscript. We have adopted the revision mode to revise the manuscript, please review it.

2. Line 25-28, In summary, this experiment proves that salinity stress can activate the HPI axis of large yellow croaker, and can also affect the immune function of large yellow croaker, and the expression of immune factors in the immune response is regulated by the upstream gene in the HPI axis.. There is an extra period at the end of the sentence.

Reply: Thanks for your valuable comments. We carefully examined the similarities in the manuscript and made revisions.

3. Line 90, what are NB, LB and HB?

Reply: Thank you very much for your question. We are very sorry that we did not provide detailed annotations for these abbreviations, which respectively represent the normal salinity group (NB), low salinity group (LB), and high salinity group (HB).

4. Line 158-160, In kidney tissue, β2-ADR was significantly up-regulated under the conditions of salinity 6, 12, 30, and 36, but there was no significant difference under the salinity of 18. why?

Reply: Thank you very much for your question. The reason why β2-ADR was not significantly upregulated in kidney tissue under the conditions of salinity 18 may be related to the osmotic pressure environment and the adaptive strategies of the organism. The salinity of 18 may be close to the physiological osmotic equilibrium point of the species, and the kidney can maintain homeostasis through basal metabolism (such as sodium potassium pumps) without activating the β2-ADR. The intensity of its salinity change has not reached the stress threshold, which is insufficient to trigger signaling cascades such as cAMP-PKA. Meanwhile, other tissues and organs may cooperate to compensate and regulate, or evolve adaptability to form a stable ion channel expression mechanism in the kidney. Thank you again for your valuable comments and hope these responses will answer your questions.

5. Line 261-262, Western Blot verified the protein expression level of the two genes CXCL12 and SOCS2. Why did you choose only these two proteins?

Reply: Thank you for your review and giving us your valuable comments, which will help us to better improve the manuscript.

CXCL12 as a core member of chemokines regulates immune cell migration and inflammatory response, and its protein expression can reflect immune activation status. While SOCS2 is a key negative regulator of the JAK-STAT pathway, maintaining immune homeostasis. The two represent the activation and inhibition mechanisms of the immune system, respectively, and can systematically reveal the bidirectional regulation of the HPI axis on immunity. In addition, both proteins have high stability and specific antibodies have been verified to be applicable, indicating strong technical feasibility. Bioinformatics predicts that CXCL12 is regulated by the β2-ADR/cAMP-CREB pathway, and SOCS2 directly inhibits IL6 signaling. As a key hub of HPI immune interaction, its protein validation can be linked to a molecular regulatory network, providing direct evidence for mechanism analysis.

Reviewer 2 Report

Comments and Suggestions for Authors

Chen et al. investigated the effects of salinity on the hpothalamic-pituitary-interrenal axis (HPI), HPI axis-related genes, and immune-related gene expression in large yellow croaker, as well as their internal regulatory mechanisms. This result showed that salinity stress can activate the HPI axis of large yellow croaker, and can also affect the immune function of large yellow croaker, and the expression of immune factors in the immune response is regulated by the upstream gene in the HPI axis. This study has practical significance for the healthy aquaculture of large yellow croaker. The manuscript needs to be improved before it is accepted. As follows:

1) Why did the author consider using salinity as a stress factor? Please explain?

2) The abstract should be more concise and clear.

3) Line 10: It should be changed to: “The large yellow croaker is one of the most popular economic fish species in China because of its rich edible value.”

4) Line 24: Delete “The”.

5) Line 26: It should be changed to: “and affect its immune function…...”

6) Line 298: “Epinephelus coioides” Latin names need to be italicized, please check the manuscript.

7) Line 324: “Transcription and expression” should be changed to “transcription and expression”.

8) Line 328-329: Delete “that is”, “responses”.

9) Line 505-506: The author should provide the dilution ratio of antibodies, as antibody concentration might affect the accuracy of protein expression results.

10) The use of certain terms in the article is inconsistent, such as "β 2-ADR" and "β 2-adrenergic receptor agonists".

11) There are some repetitive contents in the article, such as repeatedly mentioning the expression of GH and PRL in different tissues when discussing them. Suggest merging duplicate information to avoid redundancy.

12) The conclusion section seems to excessively repeat the results, without fully emphasizing the specific conclusions drawn from these results and their contributions to the relevant field.

Comments on the Quality of English Language

The language should be revised.

Author Response

Chen et al. investigated the effects of salinity on the hpothalamic-pituitary-interrenal axis (HPI), HPI axis-related genes, and immune-related gene expression in large yellow croaker, as well as their internal regulatory mechanisms. This result showed that salinity stress can activate the HPI axis of large yellow croaker, and can also affect the immune function of large yellow croaker, and the expression of immune factors in the immune response is regulated by the upstream gene in the HPI axis. This study has practical significance for the healthy aquaculture of large yellow croaker. The manuscript needs to be improved before it is accepted. As follows:

Reply: Thank you for your constructive comments on our manuscript. These questions are very helpful in improving the quality and clarity of our manuscript. We have made detailed revisions to the manuscript and checked every detail in it. The response to each of your comments are as follows. Please criticize and correct.

1. Why did the author consider using salinity as a stress factor? Please explain?

Reply: Thanks for your valuable comments. The selection of salinity as a stress factor in this study is mainly based on its physiological regulation mechanism and the dual value of aquaculture applications. Salinity changes directly disrupt the osmotic pressure balance of fish, activate the HPI axis, trigger cortisol release, and regulate ion transporters, providing an ideal model for revealing neuroendocrine immune interactions. Salinity, as a non pathogenic stressor, can eliminate microbial interference and directly correlate environmental stress with innate immune regulation mechanisms. In addition, large yellow croaker as a widely saline aquaculture fish, the fluctuation of salinity often leads to aquaculture diseases. Elucidating its molecular adaptation mechanism can provide targets for stress resistant breeding and aquaculture management (such as optimizing salinity parameters and enhancing immune homeostasis), which has important industrial application value.

2. The abstract should be more concise and clear.

Reply: Thank you for your valuable comments on our manuscript. We have revised the abstract using a revision mode to make it more concise and clear. Please review it.

3. Line 10: It should be changed to: “The large yellow croaker is one of the most popular economic fish species in China because of its rich edible value.”

Reply: Thanks for your carefully review. We have revised this sentence using a revision mode. Please review it.

4. Line 24: Delete “The”.

Reply: Thanks for your carefully review. We have revised it in the manuscript. Please review it.

5. Line 26: It should be changed to: “and affect its immune function…...”

Reply: Thanks for your carefully review. We have revised it in the manuscript. Please review it.

6. Line 298: “Epinephelus coioides” Latin names need to be italicized, please check the manuscript.

Reply: Thanks for your carefully review. We have revised it and carefully check the manuscript. Please review it.

7. Line 324: “Transcription and expression” should be changed to “transcription and expression”.

Reply: Thanks for your carefully review. We have revised it in the manuscript. Please review it.

8. Line 328-329: Delete “that is”, “responses”.

Reply: Thanks for your valuable comments. We have revised it in the manuscript. Please review it.

9. Line 505-506: The author should provide the dilution ratio of antibodies, as antibody concentration might affect the accuracy of protein expression results.

Reply: Thanks for your valuable comments. The primary antibody at 1:1000 dilution, and the secondary antibody at 1:5000 dilution. We have revised it in the manuscript. Please review it.

10. The use of certain terms in the article is inconsistent, such as "β2-ADR" and "β2-adrenergic receptor agonists".

Reply: Thanks for your valuable comments. We have revised it in the manuscript. Please review it.

11. There are some repetitive contents in the article, such as repeatedly mentioning the expression of GH and PRL in different tissues when discussing them. Suggest merging duplicate information to avoid redundancy.

Reply: Thanks for your valuable comments. We have revised it in the manuscript. Please review it.

12. The conclusion section seems to excessively repeat the results, without fully emphasizing the specific conclusions drawn from these results and their contributions to the relevant field.

Reply: Thanks for your valuable comments. We have revised it in the manuscript. Please review it.

Reviewer 3 Report

Comments and Suggestions for Authors

See attached file.

Comments on the Quality of English Language

The text requires a thorough English language review.

Author Response

In this study, cultured large yellow croaker was used as the research object, and RNA-seq technology was used to sequence the transcription group of different salinity (12, 24, 36) treatment of large yellow croaker brain tissue for 4 weeks. There were 6,787 differential expression genes in the high salt and normal groups, including 3,403 up-regulated and 3,384 down-regulated transcripts, and 5,525 differential expression genes in the low salt and normal groups, including 2,697 up-regulated and 2,558 down-regulated transcripts. In summary, this experiment proves that salinity stress can activate the HPI axis of large yellow croaker and can also affect the immune function of large yellow croaker, and the expression of immune factors in the immune response is regulated by the upstream gene in the HPI axis.

Reply: Thank you for your constructive comments on our manuscript. These questions are very helpful in improving the quality and clarity of our manuscript. We have made detailed revisions to the manuscript and checked every detail in it. The response to each of your comments are as follows. Please criticize and correct.

The script is quite complicatedly written. The evaluation was disrupted because the "Methods" section was placed after the "Results" section. It should be immediately after the introduction. The introduction generally describes fish stress responses and the effects of salinity on them. However, it does not explain why croaker was specifically chosen as the test species or why salinity fluctuations would be a problem for it. The text does not clarify what fish densities or salinities are typically found in actual croaker rearing units. The introduction or another early section of the manuscript should include a detailed description of croaker aquaculture practices, etc. including information on fish densities in different salinity conditions. The introduction does not clearly state the purpose of this study.

Reply: Thanks for your valuable comments. We are very sorry that the overall structure of the manuscript is not clear enough, which has disrupted the evaluation. Based on your valuable suggestions, we have revised the manuscript and hope to make improvements. In addition, according to the format requirements of the journal, the "Methods" and "Results" sections were disrupted. The following is the template format provided by the journal for your review.

This study chose the large yellow croaker as the research object because it is an important economically aquaculture fish species along the southeast coast of China, highly sensitive to changes in salinity, and salinity fluctuations are common stress factors in aquaculture environments. Clarifying the impact of salinity on its physiological regulation mechanism can provide a theoretical basis for optimizing the aquaculture environment of large yellow croaker.

According to existing aquaculture practices, the most suitable salinity is 28-30 ppt for large yellow croaker. The salinity of 16 is not conducive to the growth and development of large yellow croaker and has an impact on the survival rate and yield. When the salinity is below 22, the probability of fertilized eggs sinking in water is greater than floating up. Salinity levels below 16 and above 32 have an impact on the embryonic development of large yellow croaker. The salinity gradient set in this study covers extreme low salinity, conventional aquaculture, and high salinity environments to comprehensively evaluate the effects of salinity stress.

This study explored the expression changes of HPI axis and immune-related genes in various tissues of large yellow croaker under salinity stress, and analyzed the regulatory effect of HPI axis genes on the expression of downstream immune factors. The results will help to further understand the HPI axis and immune related molecular regulatory mechanisms of large yellow croaker under environmental stress, and provide reference for the selection of seawater salinity in aquaculture in the future.

As noted, the "Methods" section is in the wrong place. According to text (lines 443-445) One week later, the domesticated large yellow croaker was randomly divided into 6 experimental groups with salinity 6, 12, 18, 24, 30, and 36. The unit of salinity, presumably ppm, is missing from both this section and the abstract, where all treatment salinity levels are not mentioned. Information on fish densities and rearing conditions is missing from the description. The experiment should have included some variety of fish densities.

Reply: Thanks for your valuable comments. We according to the format requirements of the journal, the "Methods" and "Results" sections in the manuscript have been scrambled. We deeply apologize for any inconvenience caused to your review. The salinity levels (6, 12, 18, 24, 30, 36) are expressed in practical salinity units (ppt). This clarification has been added to the entire manuscript. Detailed rearing conditions, including barrel specifications (outer diameter × bottom diameter × height = 1.07 m × 0.88 m × 0.85 m, 50 fish/barrel) and environmental parameters (the aquaculture temperature is 23°C, dissolved oxygen is 6.5 mg/L, and the pH is 7.8-8.0), have been added to methods. While this study focused on salinity effects under standardized density conditions, we acknowledge your suggestion to include density variations as an important consideration for future experimental designs.

The text contains many abbreviations that are not explained (for example, line 466 DNBSEQ) and also includes incomplete definitions (for example, lines 477-479 The R software package was used for enrichment analysis, and the function with Qvalue <= 0.05 was regarded as significant enrichment, Wiki https://en.wikipedia.org/wiki/Hypergeometric_distribution.) What does this then mean? Lines 481-483. Trizol (TRIzol Reagent Invitrogen) reagent was used to extract total RNA for 0w, 2w, 4w, 6w, 8w, a total of 5 periods and 9 tissues in the low-salt group (6, 12, 18), high salt group (30, 36), normal group (24).

Reply: Thanks for your valuable comments. We are very sorry that our description was not accurate enough, and we have revised in the manuscript. "DNBSEQ" has been defined as a sequencing platform (DNA Nanoball Sequencing). The statistical method for enrichment analysis is clarified: "Hypergeometric tests with Qvalue ≤ 0.05 were performed using R software package". The wiki link was removed.

The text revised to "Total RNA was extracted from intestine, spleen, liver, kidney, muscle, gill, brain tissues in different salinity groups (low salinity of 6, 12, and 18 ppt, normal salinity of 24 ppt, high salinity of 30 and 36 ppt) using the trizol method (TRIzol Reagent Invitrogen) for 0W, 2W, 4W, 6W, 8W." These revisions ensure clarity and eliminate ambiguities. Thanks for your highlighting these critical issues.

The results are described in the text in excessive detail. They currently account for approximately 65% of the entire manuscript. At least half of the text needs to be trimmed, and only the essential results should remain. There are also too many figures (15 in total). Figures 2 and 3 are not results of this study.

Reply: Thanks for your valuable suggestions on the manuscript. We fully agree that there are redundant descriptions in the results section, and have made comprehensive revisions to the manuscript by streamlining the results section, removing redundant descriptions of non-critical data, and retaining only the core findings. Figures 2 and 3 show the KEGG annotation results of differentially expressed genes in transcriptome data, which have been removed here. Please criticize and correct me, thank you.

In lines 165-169: The expression of GH in liver, spleen and other tissues was also up-regulated, but the expression in the intestine had no significant difference with salinity changes. In addition, the expression level of GH at 2W and 4W was not significantly different from that of the control group but showed an upward-regulated trend at 6W and 8W (Figure 5). Statistical analyses that would show significant differences or trends are completely missing from the text. This issue is repeated several times later on. For example, in lines 244-250: Subsequently, the expression correlation analysis of the expression levels of the above-mentioned immune-related genes and the key genes of the HPI axis in kidney tissues showed that the correlation between CXCL12 and β2-ADR was mainly manifested as a significant correlation under low salt, while it was highly correlated with GH. IL6 is more positively correlated with β2-ADR, and negatively correlated with GH. SOCS6 has a good correlation with β2-ADR or GH, and it is highly correlated with β2-ADR. Have these statistical tests been conducted after all?

Reply: Thanks for your careful review and valuable comments. Regarding the explanation of statistical analysis in the article, we have added a significant difference describe in the revised manuscript, including p-values and significant differences. In addition, the correlation analysis results have been statistically tested before. Due to excessive redundancy in the results, these results were not included in the manuscript. We are now uploading this data in the form of supplementary materials for your review. Thank you again for your valuable review.

Supplementary Figure. Correlation between the expression of immune-related genes and HPI axis related genes in different periods and different salinities

In figures 4-14, the y-axis is labeled as "Relative mRNA expression level," but it is not defined. What is its unit, and how was it determined? Figure 15 should be completely omitted.

Reply: Thanks for your valuable comments and correction. The Y-axis in the figure represents the "relative mRNA expression level", which is a standardized calculation based on qPCR data, measured in fold change. It is determined using the 2−ΔΔCT method, with the control group as the reference, and calculated after adjusting for internal reference genes.

Figure 15 (revised manuscript as Figure 14) shows the expression results of WB protein. The protein expression data was quantified through grayscale analysis, and the grayscale values of the target protein band and the reference protein band were measured using ImageJ software. The ratio of the target protein grayscale value to the reference protein grayscale value was used as the standardized relative expression level, and the results were expressed as "relative protein expression level". The control group was used as the benchmark (set as 1), and the experimental group data was the fold change relative to the control group. Thank you again for your correction.

Supplementary Figure. Results of protein relative expression level based on gray value analysis

The experimental study did yield interesting results, the significance of which has been discussed in the "Discussion" section (e.g., lines 355-362). But what is their practical significance? Can the farming locations for croaker be changed to favor salinity levels where stress effects would be reduced? And what is the combined effect of stocking densities and seawater salinity? What would the optimal farming conditions be for croakers? What recommendations can now be made to fish farmers based on the results obtained, and what practical implications do these have?

Reply: Thanks for your valuable comments and correction. This study revealed the dynamic regulatory relationship between the HPI axis and immune genes in large yellow croaker under salinity stress. Its practical application value is reflected in the fact that long-term high salinity environment can significantly enhance the stress response of large yellow croaker (such as severe fluctuations in genes such as IL6 and CXCL12). It is recommended to optimize the salinity levels of aquaculture, or through staged salinity management (such as low salinity during seedling stage and gradually adjusting to high salinity during adult stage). In order to balance stress resistance and growth performance. Meanwhile, it is proposed to synergistically reduce aquaculture density and strengthen water quality monitoring in high salinity environments, in order to alleviate the stress superposition effect of salinity density. For the actual production process, it is recommended to use a freshwater buffer system to stabilize salinity in areas with salinity fluctuations. During the adult fish stage, immune markers such as IL6st and SOCS2 in gill, liver, and kidney tissues should be regularly detected as stress warning indicators. In areas where salinity cannot be controlled, high-density aquaculture should be prioritized during the low salinity season. These suggestions not only provide theoretical basis for precise salinity management, selection of salinity tolerant strains, and development of immunoprotective agents, but also establish an operable technical path for stress resistance regulation and disease prevention and control system in aquaculture industry. In addition, the optimal aquaculture conditions for large yellow croaker require maintaining a salinity of 25-30 ppt, avoiding short-term drastic fluctuations, and controlling environmental parameters such as water temperature at 21-25℃, dissolved oxygen ≥ 6 mg/L, and pH 7.8-8.0. The stocking density is 3-5 kg/m3. Your valuable comments greatly enhances the application value of this research. Thank you again.

The conclusions do not address these essential questions at all.

Reply: Thanks for your valuable comments. We have rewritten the conclusion, please criticize and correct it again.

This study reveals the key mechanism by which salinity stress affects the neuroen-docrine immune regulatory network of large yellow croaker. By integrating transcriptom-ics and molecular validation techniques, it was found that HPI axis-related genes (β2-ADR, GH, PRL) and immune genes (CXCL12, SOCS2) exhibit spatiotemporal specific co expression characteristics during salinity adaptation, and their interaction patterns dynamically change with stress intensity and duration. It is worth noting that bidirec-tional changes in salinity (low and high salinity) differentially regulate the protein ex-pression of CXCL12 and SOCS2, suggesting that this signaling node may serve as a mo-lecular switch in salinity stress. These results provide a new perspective for analyzing the salinity adaptation mechanism of fish. The HPI axis and immune interaction regulation model established by it can provide theoretical basis for salinity regulation strategies in aquaculture. By optimizing salinity parameters, variety selection and precise environ-mental management can be achieved, which has practical value for improving the sur-vival rate and disease resistance of large yellow croaker aquaculture.

We sincerely thank you for your carefully review and give us valuable comments, which are very helpful to improve the quality of the manuscript. We have made comprehensive revisions to the manuscript according to your suggestions. If there are still deficiencies in the revised manuscript, or if you have any new insights and suggestions. We welcome your continued criticism and correction. Thank you again for your patience and guidance. Thank you!

Round 2

Reviewer 2 Report

Comments and Suggestions for Authors

The concerns have been addressed well.

Author Response

Dear Reviewer:

Thanks for your comments concerning our manuscript entitled “Regulation of immune-related gene expression by salinity-induced HPI axis in large yellow croaker, Larimichthys crocea”. Those comments are all valuable and very helpful for revising and improving our research paper. All the comments were very important for guiding significance to our research. The comments have been studied carefully and the corrections have been made. In this manuscript, the revised contents are displayed in red font. Please kindly correct it if any mistake. Sincerely thank you!

Reviewer 3 Report

Comments and Suggestions for Authors

The authors have successfully improved the script significantly based on the given feedback and justify the background in their cover letter. 

However, I am still concerned about the presentation of the results. In particular, Figures 3, 4, and 6–13 are quite difficult to interpret. The authors should carefully reconsider whether all of them are necessary. In any case, the bar charts must be made in color. 

Figure 14 does not add value at this point and can be omitted.

Comments on the Quality of English Language

I strongly recommend a thorough language review.

Author Response

Dear reviewer:

We sincerely appreciate your valuable suggestion, which has greatly helped improve our manuscript. At the same time, we sincerely appreciate your recognition of our manuscript revisions. We will continue to improve the manuscript and strive to present higher quality papers.

We have remade the images based on your comments and uniformly represented them in color to improve readability. Due to the involvement of these genes in the HPI axis and immune related pathways, which need to be presented for the sake of research integrity and scientific significance. We hope you can understand the purpose of doing so. Meanwhile, we have revised the description of the relevant results, hoping to make readers better understand.

In accordance with your recommendation, we have removed “Figure 14” from the manuscript to streamline the presentation of our results and enhance the clarity of our findings. We believe this modification has helped to improve the overall focus and coherence of the study.

We sincerely appreciate your valuable suggestion regarding the English language quality. We have carefully revised the manuscript to improve the language quality.

Your suggestions are crucial for improving our paper, and we sincerely appreciate your meticulous and professional review comments! If you have any other questions or suggestions, we will continue to actively cooperate and make revisions.
